# Design of and Experiment on a Film Removal Device of an Arc-Toothed Residual Film Recovery Machine before Sowing

**Shuaikang Xue, Xuegeng Chen, Jingbin Li \*, Xianfei Wang and Zhiyuan Zhang**

College of Mechanical and Electrical Engineering, Shihezi University, Shihezi 832003, China; 20192009026@stu.shzu.edu.cn (S.X.); chenxg130@shzu.edu.cn (X.C.); wangxf@shzu.edu.cn (X.W.); zhangzy@stu.shzu.edu.cn (Z.Z.)
* Correspondence: Lijingbin@shzu.edu.cn

**Abstract:** In view of the serious film wrapping phenomenon and poor film removal effect of the film removal devices of residual film recovery machines, a combined "mechanical + air flow" film removal device is designed. It is mainly composed of vane-type film removal rollers and diversion shells and can complete film removal and film transportation in turn. The analysis and parameter design of the key working parts, named film stripping blades, are carried out. The condition of film removal is calculated by force analysis, and the internal flow field of the device is simulated based on the Fluent software. Taking rotating speed of the vane-type film removal roller, the inclination angle of the film stripping blade, and the diameter of the roller as test factors, and the area ratio of the vortex region to the effective region as the evaluation index, a three-factor three-level orthogonal simulation test is designed. The response surface model of each test factor is established, and the significance of each test factor on the evaluation index is analyzed. Through optimization, the optimal parameter combination suitable for the film removal flow field is obtained as follows: the rotating speed of the vane-type film removal roller is 283 r/min, the inclination angle of the film stripping blade is 25° and the diameter of the roller is 219 mm. Under the optimal combination of parameters, the device is manufactured, and the effect of the device is verified by a field test. The results show that the film removal rate of the device is 98.04%, and there is no film wrapping phenomenon in the operation process, which can meet the needs of residual film recovery before sowing.

**Keywords:** residual film recovery; combination type; film removal device; design; test

## 1. Introduction

Plastic film mulching is an important agricultural production technology [1] that has the functions of maintaining fertilizer, increasing temperature and reducing crop diseases [2]. However, with the wide application of plastic film, plastic film that has not been recycled in time changes the physical structure of the soil, hinders the normal transportation of soil water and nutrients, and even causes crop yield reduction and destroys sustainable agriculture production [3,4]. In the face of the increasingly serious problem of residual film (residual film can be explained as mulch film that was not been recovered in time in farmland), the Ministry of Agriculture and Rural Affairs and others jointly issued "opinions on accelerating the prevention and control of agricultural film pollution" in 2019, and the Ministry of Agriculture and Rural Affairs again launched "management measures for agricultural film" in 2020 [5,6]. The problem of residual film pollution is highly valued by government departments and urgently needs to be solved [7].

In foreign countries, the plastic film thickness is greater than 0.020 mm and the film has strong damage resistance [8], so it is easy to recover. However, the plastic film thickness used in China is generally less than 0.008 mm, which can easily be torn in the process of mechanical recovery [9], the recovery of residual film is a difficult problem. At this stage, the methods of residual film recovery include manual recovery, the use of degradable film, and mechanized recovery [10]. Among them, the cost of manual recovery is too high; As

degradable film technology is not particularly mature and difficult to process [11], it is not suitable for China's national conditions. Therefore, mechanized recovery is the main trend at present [12].

The recovery of residual film before sowing is the main recovery method [13]. The residual film is mostly fragmented and distributed on the surface and shallow soil, so the recovery of residual film is difficult, and the recovery effect is not ideal [14,15]. To solve the problem of residual film recovery before sowing, China has developed a variety of residual film recovery machines [16]. The film removal device is not only the core working part of the residual film recovery machine, but also the main difficulty in the development of residual film recovery machines [17]. At this stage, there are relatively few studies on film removal devices, mainly including the pneumatic type [18], telescopic rod tooth type [19] and stripping blade type [20]. The structure of the pneumatic stripping device is relatively simple. The main working component is the fan. During operation, the film removal mainly depends on the force of the air flow on the residual film, but the film removal effect is not good [21,22]; the telescopic rod tooth removal device mainly depends on the expansion and contraction of the elastic teeth on the eccentric roller to pick up and remove the film. It has high development cost and low reliability [23]; the stripping blade type removal device mainly adopts a method in which the rotating direction of the working part of the film removal device is the same as that of the film picking device, the film removal rate is not high and the film wrapping phenomenon easily occurs on the film stripping blades [24].

Addressing the problems of a low film removal rate and serious film wrapping phenomenon in the film removal device of residual film recovery machines before sowing, this paper designs a film removal device composed of a vane type film removal roller and a diversion shell, according to the film picking principle of an arc-toothed residual film recovery machine. The film removal device combines the advantages of mechanical stripping and pneumatic stripping. Therefore, it is named "Mechanical + airflow" combined film removal device [25]. Through theoretical analysis, a film removal blade, which is the key working part of a film removal device, is designed, and its parameters are determined. This paper uses the Fluent software (high efficiency, low cost, and strong repeatability) to simulate the internal flow field of a film removal device, explores the influence of different structures and working parameters of the vane-type film removal roller on the internal flow field of the device, obtains the optimal combination of flow field parameters suitable for the operation of the film removal device, and performs verification tests. It is expected that the film removal device can smoothly complete residual film removal and transportation, which can provide a reference for research on film removal devices for residual film recovery machines.

## 2. Materials and Methods

### 2.1. Structure and Working Principle of the Whole Machine

The arc toothed residual film recovery machine is mainly composed of a film picking device, film removal device, and residual film recovery box, as shown in Figure 1. The whole machine is connected to the tractor through a three-point suspension device. The power of the rear output shaft of the tractor is first transmitted to the chain transmission mechanism on both sides of the frame through the gearbox, and then the chain transmission mechanism drives the film picking device and the film removal device. During the operation, first, with the advance of the tractor, the arc-shaped teeth rotate anticlockwise to go deep into the shallow soil and pick up the residual film. Second, when the residual film on the arc-shaped teeth moves to the film removal area, it contacts the high-speed rotating film stripping blades, and the film stripping blades have the effect of "scraping" and "hitting" the residual film to complete the film removal work [26]. Finally, the residual film is transported into the residual film recovery box by the airflow generated by the film stripping blades in the film removal device.

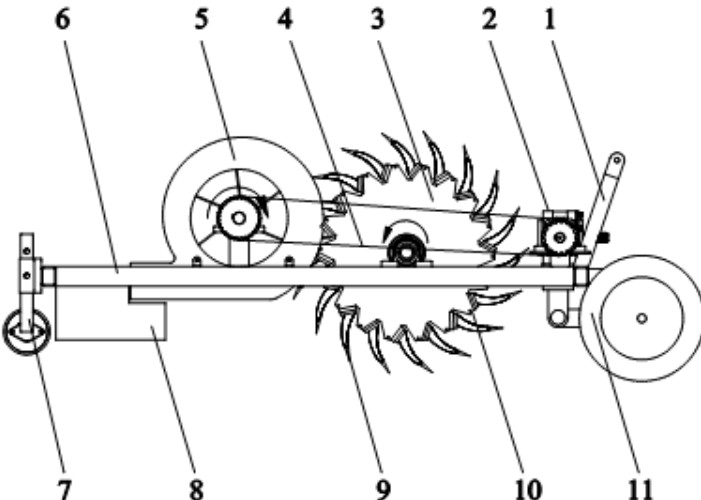

**Figure 1.** Structure diagram of the whole machine. 1. Three-point suspension device. 2. Gearbox. 3. Film picking device. 4. Chain drive mechanism. 5. Film stripping device. 6. Frame. 7. Suppressing soil cover device. 8. Residual film recovery box. 9. Arc-shaped tooth. 10. Spoke plate. 11. Depth limit wheel.

### 2.2. Design of Film Removal Device

As shown in Figure 2, the film removal device is mainly composed of a vane-type film removal roller 5 and diversion shell 4. The vane-type film removal roller 5 includes a roller 10, film stripping blade 15 (hereinafter referred to as blade), and blade connecting plate 11. The roller 10 material is a seamless steel pipe with a certain wall thickness. The blade connecting plate 11 is evenly welded on the surface of the roller 10. The main structure of the diversion shell 4 is composed of the top cover 3, the bottom plate 7 and the side plate 2. One end of the diversion shell 4 is provided with a film removal area 6, which is convenient for the interaction between the arc-shaped tooth and the blade 15. The other end is provided with a film conveying port 9 to guide the residual film into the residual film recovery box.

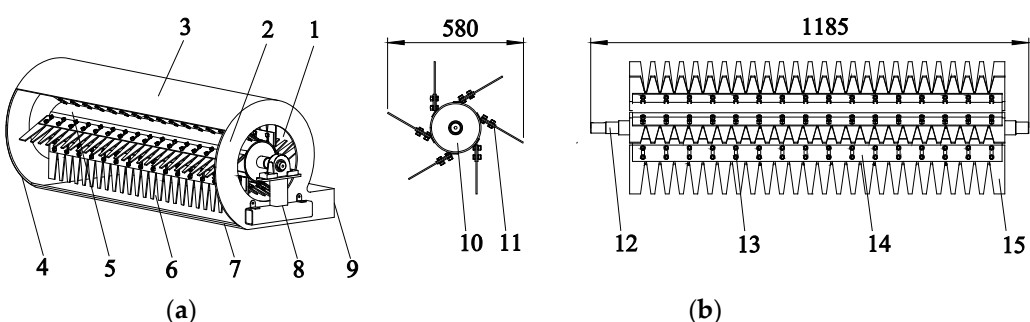

**Figure 2.** Structural schematic diagram of the film removal device (**a**) and structure diagram of the vane-type film removal roller (**b**). 1. Air inlet. 2. Side plate. 3. Top cover. 4. Diversion shell. 5. Vane type film removal roller. 6. Film removal area. 7. Bottom plate. 8. Frame. 9. Film conveying port. 10. Roller. 11. Blade connecting plate. 12. Shaft. 13. Bolt. 14. Blade pressing plate. 15. Blade.

The film removal device is the core part of an arc-toothed residual film recovery machine, and it is also the key to determining the film removal rate. As the residual film before sowing is mostly in the form of a "broken", soft texture and strong electrostatic adsorption force, to prevent tearing of the residual film during film removal, the rotation direction of the vane-type film removal roller is designed to be opposite to that of the film picking device [27,28], as shown in Figure 1. The working principle of the film removal device is that the blade acts on the residual film on the same arc-shaped tooth many times.

After overcoming the winding and adsorption force of the residual film on the arc-shaped tooth, the residual film is removed, that is, mechanically stripped. When the vane-type film removal roller rotates at high speed in the diversion shell, a stable air flow field is generated inside the diversion shell. The removed residual film is blown into the residual film recovery box along the direction of the film conveying port by virtue of the air flow force, that is, airflow-type film removal, which reduces the winding phenomenon between the removed residual film and the blade.

### 2.3. Design of Blade

The blade is the key part of the film removal device, which needs to interact with the arc-shaped tooth frequently. Its material is 8 mm thick rubber belt. Compared with other rigid materials, rubber belt has good elasticity [29]. When the blade is in elastic contact with the arc-shaped teeth, the wear of the arc-shaped teeth can be reduced [30]. In addition, the rubber belt has a higher friction coefficient, which can well grasp the residual film on the arc-shaped teeth and has a better film removal effect [31].

#### 2.3.1. Design of Tooth Profile of Blade

The design of the tooth profile of the blade has a direct impact on the film removal effect, which is based on the structural parameters and arrangement of the arc-shaped teeth. As shown in Figure 3, the design of the tooth profile of the blade includes the determination of three parameters $a$, $b$ and $c$. The tooth profile of the blade is designed as an isosceles trapezoid. The arc-shaped teeth are installed in a staggered arrangement, and the distance between the adjacent arc-shaped teeth is 60 mm. For $c$, $c$ is the distance between adjacent teeth. According to the working principle of film removal, the arc-shaped tooth needs to pass through the adjacent teeth of the blade. It is necessary to ensure the distance between the adjacent arc-shaped teeth such as the distance between adjacent teeth. Therefore, $c$ should be 60 mm. For $b$, $b$ is the tooth height of the blade. In the process of film removal, the tooth height of the blade should ensure that the blade can reach the area where the residual film is located. As in the pretest of the film picking device, the research group found that the residual film was mainly concentrated approximately 80 mm from the arc-shaped tooth tip in the vertical direction, and the width of the arc-shaped tooth at this position was 25 mm. Therefore, $b \geq 80$ mm and when $b$ is 120 mm, the requirement of film removal can be met. For $a$, $a$ is the maximum distance between the teeth of adjacent blades. When the value of $a$ is too large, the arc-shaped tooth passes through the blade and cannot play the role of film removal, so $a \leq 25$ mm. If the value of $a$ is too small, the blade deforms greatly, and the force on the residual film on the arc-shaped tooth is large during the operation, which is conducive to film removal, but easily causes residual film entrainment on the film removal blades. Therefore, to sum up, $a = 20$ mm is suitable for the design.

#### 2.3.2. Determination of the Number and Inclination Angle of the Blade

The number and inclination angle of the blades are the key structural parameters of the film removal device. The former design mainly considers two aspects: first, the effect of film removal. When the speed of the vane-type film removal roller is fixed, the greater the number of blades is, the greater the contact frequency between the blades and arc-shaped teeth is. It is necessary to ensure that the same arc-shaped teeth are acted by blades at least twice when the film picking device rotates one full circle. Second, dynamic balance. The blades are arranged at equal angles along the tangential direction of the surface of the roller, and the axial direction is symmetrical, which is conducive to the dynamic balance of the vane-type film removal roller [32]. Considering the above factors and combined with the structure of the whole machine, the number of blades was determined to be 6.

There are three ways to install the blade: forward tilt, radial and backward tilt [33]. As shown in Figure 4, $F_1$ is the force of the blade on the residual film when blades interact with arc-shaped teeth. Decompose $F_1$ along the tip of the arc-shaped tooth and its normal direction to obtain $F_2$ and $F_3$. The main function of $F_3$ is to make the residual film move

along the direction of the tip of the arc-shaped tooth. At the same time, it exerts the effective force of the blade on the residual film, and $F_3 = F_1 \cos \beta$. When $F_1$ is fixed, for different blade installation methods, the size of $F_3$ is as follows: forward tilt > radial > backward tilt. When a blade is installed with a forward inclination and an inclination of 15° (the blade is at the dotted line position in Figure 4), the blade is perpendicular to the tip of the arc-shaped tooth. At this moment, the effective force of the blade on the residual film reaches the maximum value, which is conducive to the removal of the residual film. Therefore, to better explore the effect of the blade inclination on the film removal effect, blades are evenly distributed on the surface of the roller by forward tilt, and the range of tilting angles is 5° to 25°.

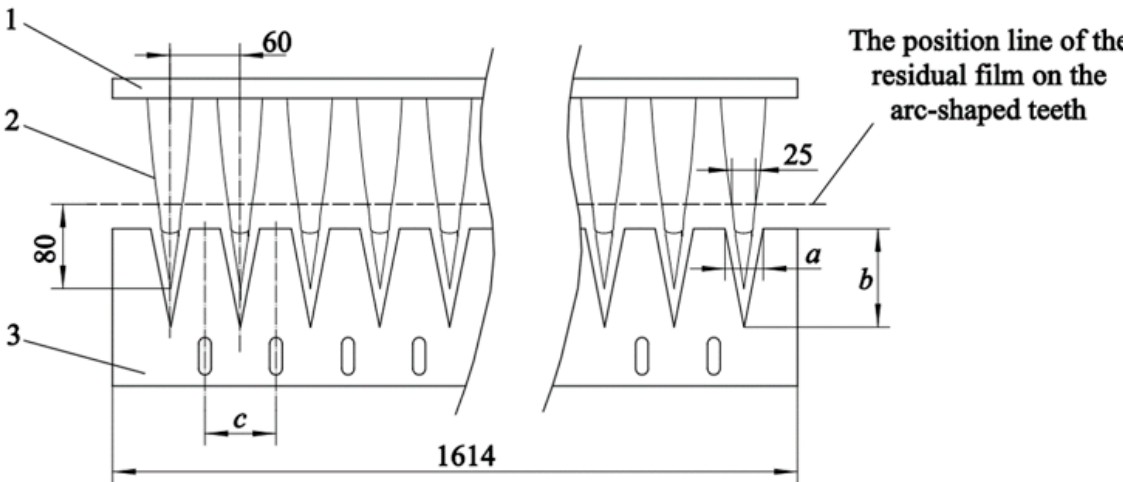

**Figure 3.** Schematic diagram of the working of blade and arc-shaped tooth. 1. Spoke plate. 2. Arc-shaped tooth. 3. Blade. Note: *a* is the maximum distance between the teeth of adjacent blades, mm; *b* is the tooth height of the blade, mm; *c* is the distance between adjacent teeth, mm.

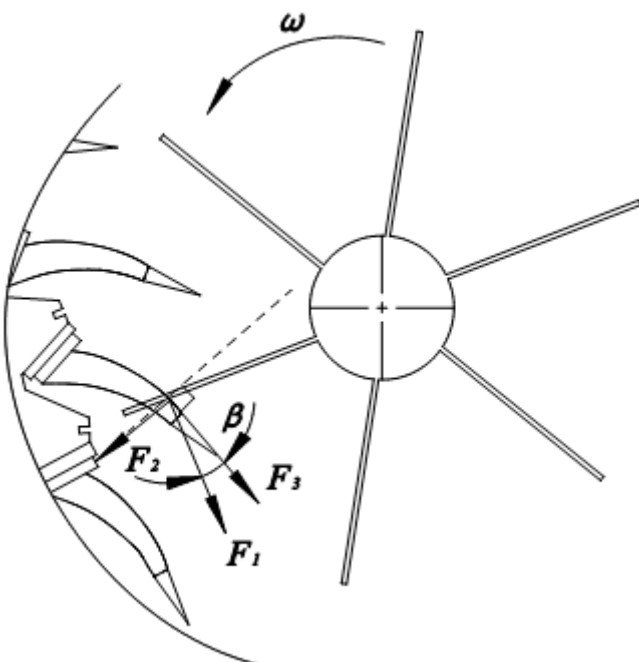

**Figure 4.** Schematic diagram of the force of the blade on the residual film.

### 2.3.3. Analysis of the Conditions of Film Removal

To separate the residual film from the blade, it is necessary to ensure that the centrifugal force of the residual film at the end of the blade is greater than the adsorption and winding force between them [34]. Due to the easy adsorption and floating of the residual film, the resistance effect of airflow should be considered in the motion analysis of the residual film [35]. Figure 5 shows the instantaneous force analysis of residual film separation.

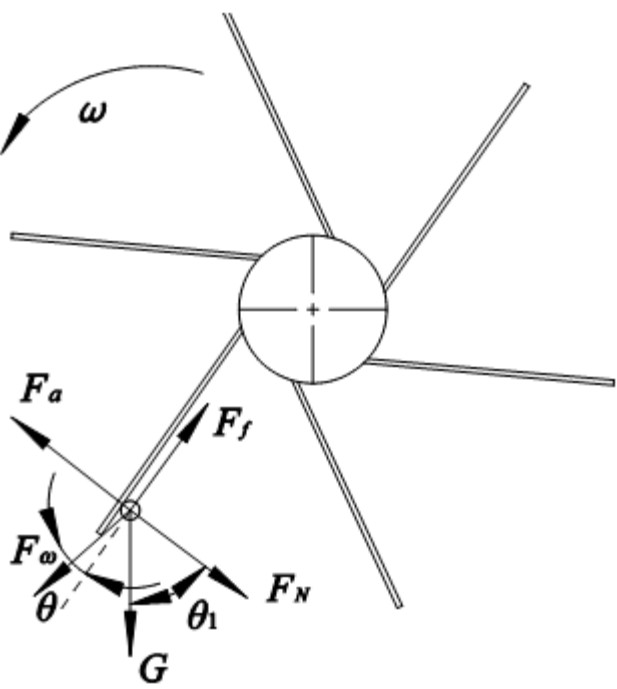

**Figure 5.** Schematic diagram of the force analysis of the separation of the residual film and blade. Note: $F_a$ is the air resistance of the residual film, N; $F_\omega$ is the centrifugal force on the residual film, N; $G$ is the gravity of the residual film, N; $F_N$ is the supporting force of blade to residual film, N; $F_f$ is the friction force of the residual film, N; $\theta$ is the inclination angle of the detached blade, (°); $\theta_1$ is the angle between the gravity of the residual film and the vertical direction of the detached blade, (°).

In the separation process of the residual film, according to the force balance relationship, the following results can be obtained:

$$\begin{cases} F_\omega \cos\theta + G\sin\theta_1 = F_f \\ G\cos\theta_1 + F_N = F_a + F_\omega\sin\theta \\ F_a = 0.5c1psv_1^2 \end{cases} \tag{1}$$

Note: $c1$ is the resistance coefficient, and the value 0.4 is used in the calculation [36]; $p$ is the air density, 1.29 kg/m$^3$; $s$ is the windward area of the residual film, m$^2$; and $v_1$ is the linear velocity of the blade, m/s.

According to Equation (1), the conditions of film removal are as follows:

$$F_\omega \cos\theta + G\sin\theta_1 > F_f \tag{2}$$

Namely:

$$v_1 < \sqrt{\frac{mgr\sin\theta_1(1+\mu)}{\left[\mu(\frac{1}{2}cpsr + m\cos\theta) - m\cos\theta\right]}} \tag{3}$$

Note: $m$ is the mass of the residual film, kg; $\mu$ is the static friction coefficient of the detached blade, which is 0.62 [37]; and $r$ is the distance between the residual film and the rotating center of the vane-type film removal roller, m.

Moreover, according to the kinetic energy theorem, in the interaction process of the residual film, arc-shaped tooth and blade, there are:

$$F_b = \frac{m(v_1^2 - v^2)}{2L_1} \tag{4}$$

Note: $v$ is the linear velocity of the arc-shaped tooth, m/s; $L_1$ is the distance of film removal, m; and $F_b$ is the average force of the blade to the residual film on the arc-shaped tooth, N.

Equation (4) shows that the greater the linear velocity of the blade is, the greater the force of the blade acting on the residual film on the arc-shaped tooth, that is, the more beneficial it is to film removal. At the same time, because the inclination angle of the blade is designed to be forward, when the rotation speed of the vane-type film removal roller is too high, the residual film on the blade is self-locking [38]. According to many tests, it is found that when the speed of the vane-type film removal roller is lower than 100 r/min, the airflow speed inside the film removal device is lower than the suspension speed of the residual film and cotton rod in the soil, and the residual film cannot be effectively separated and transported. When the speed of the vane-type film removal roller is higher than 300 r/min, the power consumption of the device will be increased. The test shows that when the rotating speed of the blade type film removal roller is 100 r/min~300 r/min, the effect of separating the residual film from the film removal blade is better. Therefore, the rotating speed range of the blade type film removal roller is determined as 100 r/min~300 r/min.

### 2.4. Simulation Model and Setting of Simulation Parameters

The internal flow field of the film removal device has an important influence on the film removal effect, which directly determines whether the residual film removed from the arc-shaped tooth winds with the blade and whether the residual film removed can enter the residual film recovery box smoothly. Therefore, this paper uses the ANSYS Fluent19.0 to simulate the flow field inside the film removal device and use Fluent19.0 pre-processing software Ansys meshing to mesh the simulation model. The mesh type selects unstructured tetrahedral mesh. A section view of the effect of mesh dividing is shown in Figure 6. The reasonably simplified simulation model is shown in Figures 7 and 8, and the main parameters of the model are shown in Table 1.

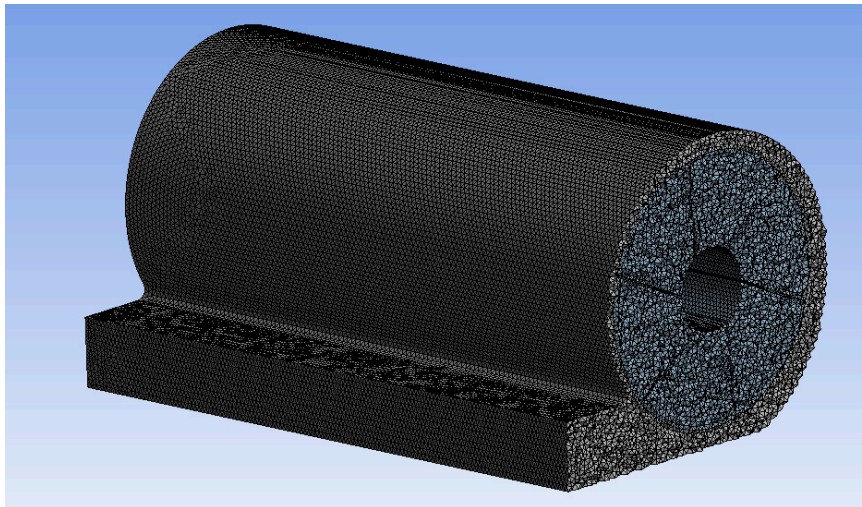

**Figure 6.** Section view of effect of mesh dividing.

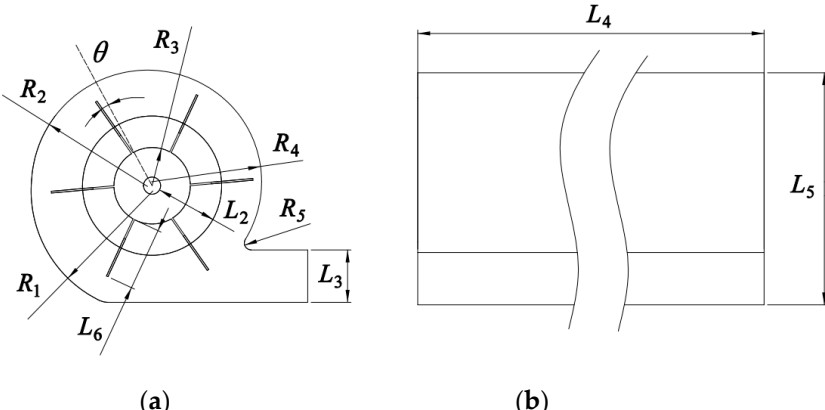

(**a**)                                (**b**)

**Figure 7.** 2D schematic diagram of simplified model. Front view (**a**) and vertical view (**b**).

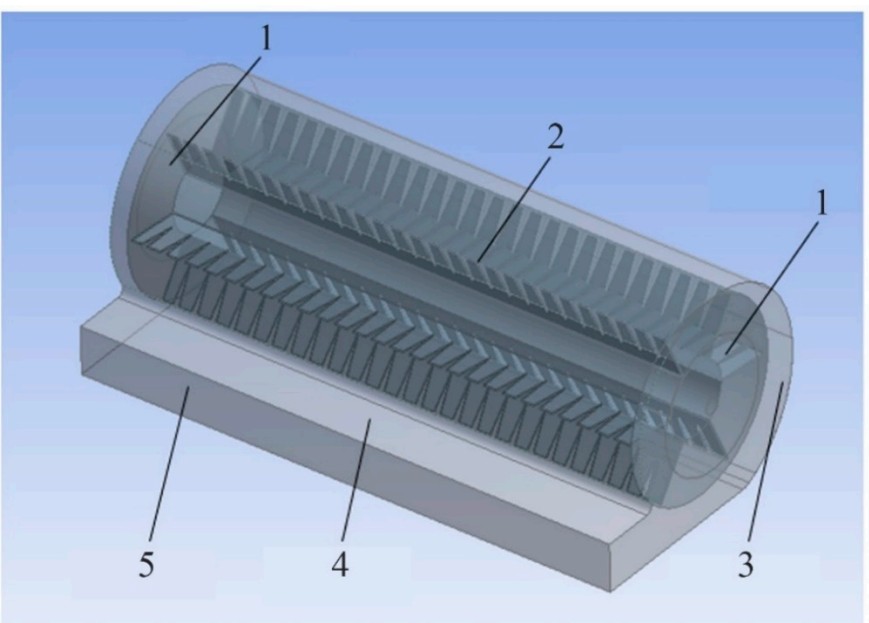

**Figure 8.** Schematic diagram of simplified model. 1. Inlet. 2. Rotating area. 3. The surface of film removal area. 4. Static area. 5. The surface of film conveying port.

**Table 1.** The main parameters of the simplified model.

| Parameters | Value |
| --- | --- |
| Curve radius of diversion shell $R_1$, $R_2$, $R_4$, $R_5$, | 349 mm, 333 mm, 316 mm, 20 mm |
| The radius of roller $R_3$ | 109.5 mm |
| The inclination angle of the blade $\theta$ | 5° |
| The size of inlet $L_2$ | 175 mm |
| The height of film conveying port $L_3$ | 150 mm |
| The length of simplified model $L_4$ | 1700 mm |
| The width of simplified model $L_5$ | 665 mm |
| The length of blade $L_6$ | 180 mm |

The relevant settings of the simulation process are as follows: the multiple reference system model is selected, the moving area of the vane type film removal roller is defined as the rotating area, the rotation speed of the vane type film removal roller is given as 200 r/min, and the remaining area of the model is defined as the static area. To better show the phenomena of airflow and vortex inside the film removal device, the Realizable k-ε model is used [39]. The pressure–velocity coupling method uses the SIMPLE algorithm

(Semi-Implicit Method for Pressure Linked Equations) [40,41], and the momentum equation, turbulent kinetic energy and other discrete formats are all used in the second-order upward style. The inlets on both sides are set as pressure inlets, the surface of the film removal area and the surface of the film conveying port are set as the pressure outlets, and the reference pressure is set to standard atmospheric pressure.

*2.5. Test Verification*

2.5.1. Test Condition

To ensure the accuracy of the test, the residual film and the mixture of film and impurities (broken cotton stalk) collected during the investigation were randomly laid on the surface of the soil and a shallow layer of soil with a depth of 0–100 mm. The test field was artificially simulated based on the data from the previous field survey to recover the environment of the residual film before sowing and the parameter settings of the test field are shown in Table 2. The test equipment included a Tiantuo TN954 tractor (Tianjin Tractor Manufacturing Co., Ltd, Tianjin, China), an electronic scale (model: JE3001, Shanghai Shengke Instrument Equipment Co., Ltd, Shanghai, China), a soil moisture meter (model: TDR300, Beijing Bolun Jingwei Technology Development Co., Ltd., Beijing, China), and a tape measure.

**Table 2.** The parameter settings of the test field.

| Parameters | Value |
|---|---|
| The average firmness of the soil (depth: 0~100 mm) | 171 kPa |
| The average moisture content of the soil (depth: 0~100 mm) | 13% |
| Residual film content in soil (depth: 0~100 mm, area: 1 m$^2$) | 7 g |

2.5.2. Test Methods

This paper carries out the test, according to the national standard GB/T 25412-2010 "Remain mulch film recovery machine" [42]. The test was repeated three times in the test field to ensure the accuracy of the data. After each test, the residual film in the residual film recovery box was cleaned, dried, and weighed. The evaluation index of this test verification selects the film removal rate, and the formula is as follows [43,44]:

$$K = \frac{m_1}{m_1 + m_2} \times 100\%$$

Note: $K$ is the film removal rate, %. $m_1$ is the weight of residual film in the residual film recovery box, g; $m_2$ is the quality of the residual film that is picked up, but does not enter the residual film recovery box, g.

**3. Results**

*3.1. Analysis of Simulation Results*

Four cross sections are selected, as shown in Figure 9, and the flow field inside the device is analyzed from the simulation results of the cross sections. Compared with the velocity vector diagram of cross sections 1~3, there is no vortex or inward backflow in the volute tongue area of the device, so the residual film is not affected by the vortex when it moves to this area and is even sucked into the diversion shell again. By comparing Figures 10b, 11b and 12b, it can be seen that: Under the action of the centrifugal force, the gas obtains kinetic energy, and the air velocity increases gradually from the inside to the outside of the blade. At the film conveying port, the air velocity on the right side is lower than that on the left side, and the velocity gradient changes obviously. In the simulation results of cross section 3 and cross section 2, the gas molecular flow and gas velocity are the same because the structure of the film removal device is symmetric about cross section 1. Figure 11a shows that a part of the gas is discharged outwards along the surface of the film removal area on the central section. However, due to the high speed of the blade, when the

residual film is separated from the blade, the rotation angle of the blade exceeds the range of the surface of the film removal area, and the residual film is not blown out of the body from this area.

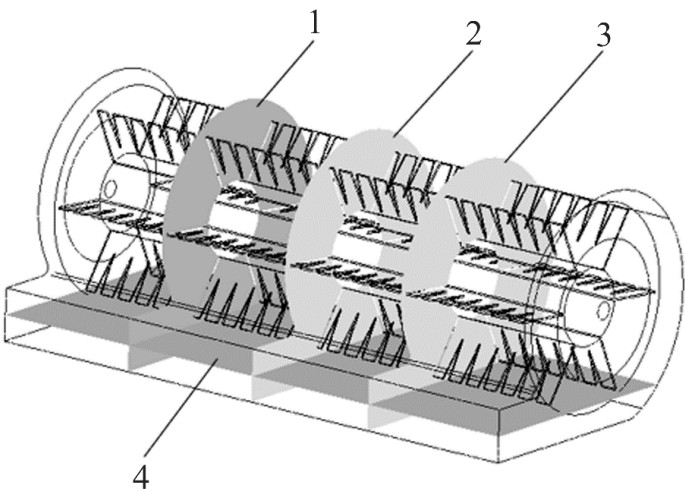

**Figure 9.** Schematic diagram of cross-section selection.1. Cross section 3. 2. Cross section 1. 3. Cross section 2. 4. Cross section 4.

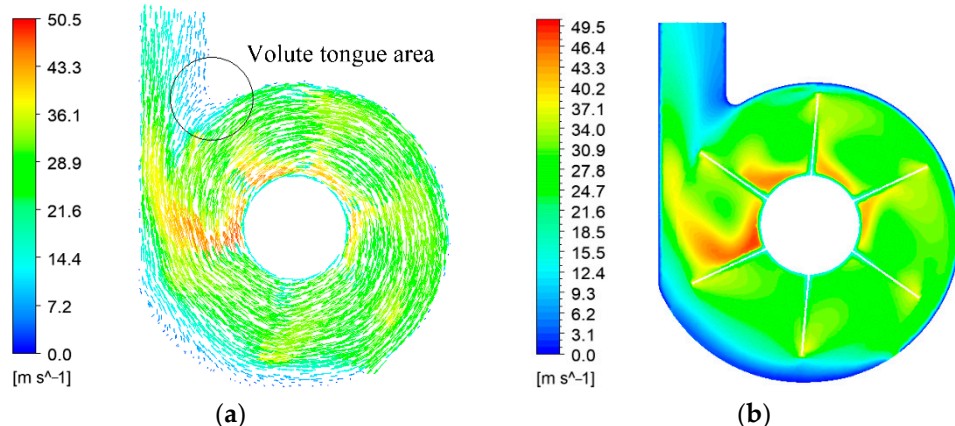

**Figure 10.** Velocity vector diagram of cross section 3 (**a**) and velocity nephogram of cross section 3 (**b**).

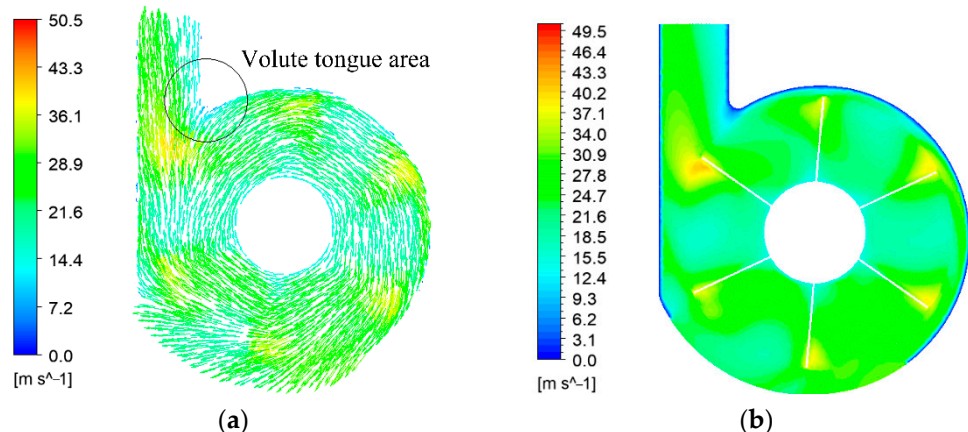

**Figure 11.** Velocity vector diagram of cross section 1 (**a**) and velocity nephogram of cross section 1 (**b**).

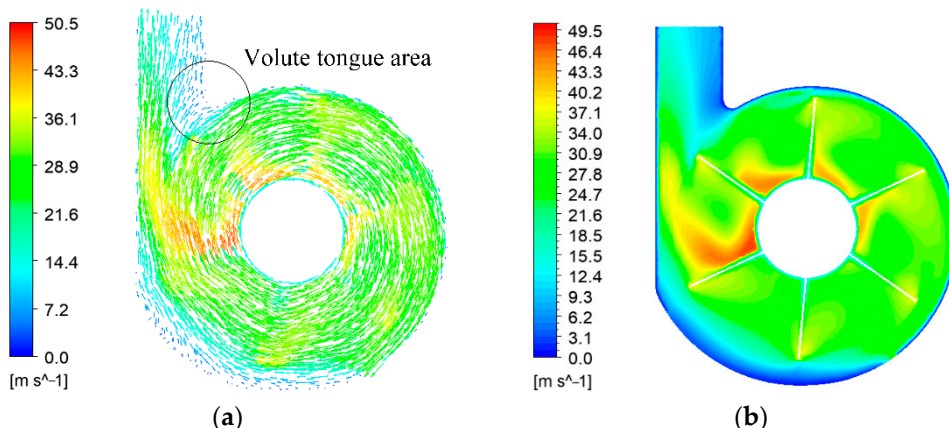

(a)　　　　　　　　　　(b)

**Figure 12.** Velocity vector diagram of cross section 2 (**a**) and velocity nephogram of cross section 2 (**b**).

Moreover, in Figure 13, the maximum velocity is 28.06 m/s, and the average velocity is 12.34 m/s. It is known that there is a small amount of cotton stalks on the surface of residual film, and the suspension velocities of residual film and cotton stalks are approximately 3 m/s and 10 m/s, respectively [45], so the gas velocity at the film conveying port meets the requirements of film removal.

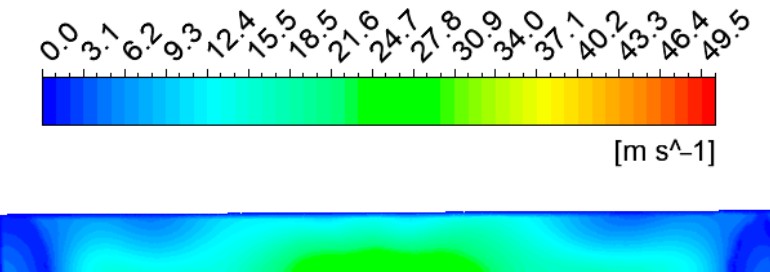

**Figure 13.** Velocity nephogram of film conveying port surface.

However, Figure 14 shows that there is a vortex region (including a small amount of inward air supply area) near the position of the cross section near the plates on both sides of the diversion shell, and the vortex region accounts for a large proportion of the effective region (the effective region is defined as the rectangular region away from the film delivery port when the film removal blade moves to the lowest point, and there is no vortex outside the effective region). The existence of the vortex region makes part of the residual film and its mixture churn and rotate violently in the diversion shell and cannot enter the residual film recovery box through the film conveying port under the action of airflow, which easily causes blockage. In severe cases, the residual film that has been removed is entangled with the blade again because of the effect of the vortex. The vortex region not only hinders the transport of residual film in the diversion shell, but also leads to a decrease in the film removal rate.

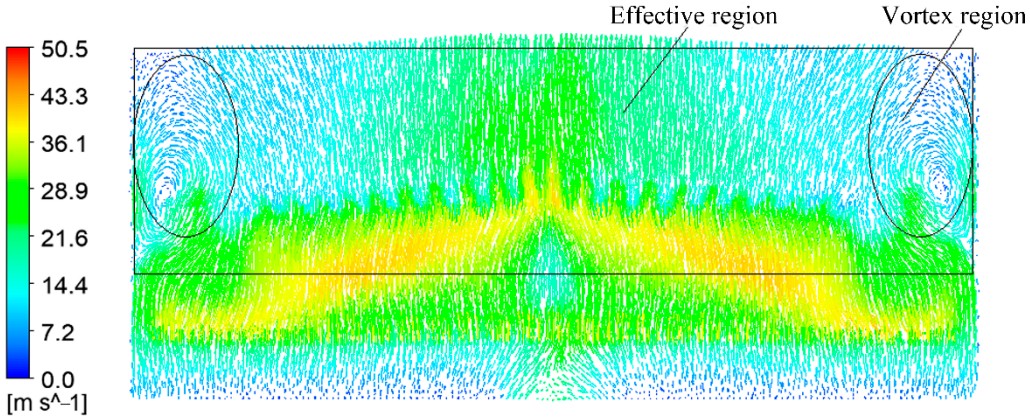

**Figure 14.** Velocity vector diagram of cross section 4.

*3.2. Design and Analysis of Orthogonal Test*

The existence of the vortex region directly determines the effect of film removal and film transportation in the device and has important research significance for the performance of the device. The rotation speed of the vane-type film removal roller, the inclination angle of the blade and the diameter of the roller affect the size of the vortex region through multiple simulations. Therefore, with the help of the Design-Expert software [46], this paper designs three factors and three-level orthogonal simulation tests for the flow field of the film removal device to solve the influence of the vortex region.

The rotating speed of the vane-type film removal roller, the inclination of the blade and the diameter of the roller are selected as test factors. As the processing material of the roller is a seamless steel tube, according to the standard size of the seamless steel tube and the processing conditions of the factory, in the test, the diameter range of the roller is selected from 140 mm to 219 mm. In addition, to reasonably reflect the influence of the vortex region, the test selects the area ratio of the vortex region to the effective region in the velocity vector diagram of section 4 (hereinafter referred to as the area ratio) as the evaluation index. The area ratio is calculated by the pixel method [47], that is, the area ratio is the ratio of the number of pixels contained in the vortex region to the number of pixels in the effective region. The smaller the area ratio is, the more suitable it is for film removal. The specific test factors and levels are shown in Table 3, and the test plan and result are shown in Table 4.

**Table 3.** Test factors and levels.

| Levels | Test Factors | | |
|:---:|:---:|:---:|:---:|
| | **Rotating Speed of the Vane Type Film Removal Roller $X_1$/(rad·min$^{-1}$)** | **Inclination Angle of the Blade $X_2$/(°)** | **Diameter of the Roller $X_3$/(mm)** |
| −1 | 100 | 5 | 140 |
| 0 | 200 | 15 | 179.5 |
| 1 | 300 | 25 | 219 |

3.2.1. Establishment of the Regression Equation and Significance Analysis of the Model

As shown in Equation (5), using the Design-Expert software to perform multiple regression fitting analysis on the test data in Table 4 [35], a regression equation with the evaluation index as the response function and 3 test factors as independent variables is obtained. Then, this paper performs an analysis of variance on the model, and the results are shown in Table 5.

**Table 4.** Test plan and result.

| Test Number | Rotating Speed of the Vane Type Film Removal Roller $x_1$ | Inclination Angle of the Blade $x_{2.5}$ | Diameter of the Roller $x_3$ | Evaluation Index |
|---|---|---|---|---|
| | | | | Area Ratio $Y$/% |
| 1 | −1 | −1 | 0 | 28.04 |
| 2 | 1 | −1 | 0 | 12.61 |
| 3 | −1 | 1 | 0 | 19.31 |
| 4 | 1 | 1 | 0 | 7.29 |
| 5 | −1 | 0 | −1 | 23.65 |
| 6 | 1 | 0 | −1 | 13.34 |
| 7 | −1 | 0 | 1 | 24.19 |
| 8 | 1 | 0 | 1 | 10.12 |
| 9 | 0 | −1 | −1 | 18.65 |
| 10 | 0 | 1 | −1 | 10.87 |
| 11 | 0 | −1 | 1 | 16.36 |
| 12 | 0 | 1 | 1 | 10.84 |
| 13 | 0 | 0 | 0 | 12.09 |
| 14 | 0 | 0 | 0 | 12.52 |
| 15 | 0 | 0 | 0 | 13.32 |
| 16 | 0 | 0 | 0 | 12.56 |
| 17 | 0 | 0 | 0 | 12.38 |

**Table 5.** Analysis of variance of regression model.

| Source of Variance | Area Ratio $Y$/% | | | |
|---|---|---|---|---|
| | Sum of Squares | Degree of Freedom | F Value | *p* Value |
| Model | 516.74 | 9 | 191.74 | <0.0001 ** |
| $X_1$ | 335.79 | 1 | 1121.37 | <0.0001 ** |
| $X_2$ | 93.50 | 1 | 312.25 | <0.0001 ** |
| $X_3$ | 3.13 | 1 | 10.44 | 0.0144 * |
| $X_1 X_2$ | 2.91 | 1 | 9.71 | 0.0169 * |
| $X_1 X_3$ | 3.53 | 1 | 11.80 | 0.0109 * |
| $X_2 X_3$ | 1.28 | 1 | 4.26 | 0.0778 |
| $X_1^2$ | 65.42 | 1 | 218.47 | <0.0001 ** |
| $X_2^2$ | 0.37 | 1 | 1.24 | 0.3026 |
| $X_3^2$ | 7.22 | 1 | 24.10 | 0.0017 ** |
| Residual | 2.10 | 7 | | |
| Lack of fit | 1.26 | 3 | 2.03 | 0.2526 |
| Pure error | 0.83 | 4 | | |
| Cor total | 518.84 | 16 | | |

Note: *p* < 0.01 (very significant, **); *p* < 0.05 (significant, *).

The results show that in the aspect of the significant influence of test factors on the evaluation index, the rotating speed of the vane type film removal roller is larger than the inclination angle of blade, which is larger than the diameter of the roller. The *p*-value of the model is less than 0.0001, which indicates that the regression model is extremely significant. The lack of fit item $p = 0.2526 > 0.05$ shows that the model has a high degree of fit and has good practical research significance.

$$Y = 12.57 - 6.48X_1 - 3.42X_2 - 0.62X_3 + 0.85X_1X_2 - 0.94X_1X_3 + 0.57X_2X_3 + 3.94X_1^2 + 0.30X_2^2 + 1.31X_3^2 \tag{5}$$

### 3.2.2. Analysis of the Influence of Test Factors on the Evaluation Index

Figure 15a is the response surface diagram of the interaction between the inclination angle of the blade and the rotation speed of the vane-type film removal roller to the area ratio when the diameter of the roller is at the zero level (179.5 mm). Figure 15a shows that in the direction of the inclination angle of the blade, the area ratio gradually decreases as

the inclination angle of the blade increases. In the direction of the rotation speed of the vane-type film removal roller, the area ratio first decreases with increasing rotation speed of the vane-type film removal roller, and then the range of change tends to be flat. The response surface changes greatly along the direction of the rotation speed of the vane-type film removal roller. In addition, at the zero level, compared with the effect of the inclination angle of the blade on the area ratio, the effect of the rotation speed of the vane-type film removal roller on the area ratio is more significant.

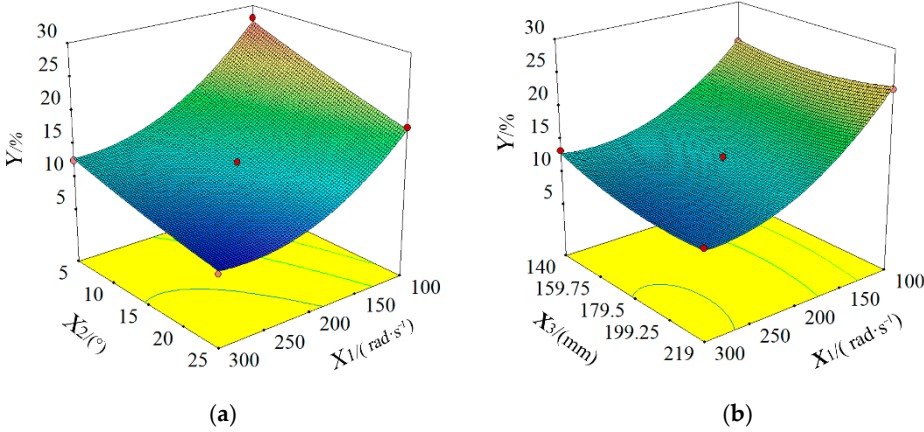

**Figure 15.** The influence of the inclination angle of the blade and the rotating speed of the vane-type film removal roller on the area ratio (**a**) and the influence of the diameter of the roller and the rotating speed of the vane-type film removal roller on the area ratio (**b**).

The main reasons for the above phenomenon are as follows: When the diameter of the roller is fixed, the space that can hold the gas in the film removal device remains unchanged. With the increase in the rotating speed of the vane-type film removal roller, the air inflow and pressure in the film removal device increase, and the phenomenon of inward air supply on the film conveying port surface decreases. At the same time, with the increase in the inclination angle of the blade, the gas distribution in the axial direction of the device is more balanced, and it is helpful to reduce the size of the vortex region at both ends of the device.

Figure 15b is the response surface diagram of the interaction between the diameter of the roller and the rotation speed of the vane-type film removal roller to the area ratio when the inclination angle of the blade is at the zero level (15°). Figure 15b shows that with increasing roller diameter and rotating speed of the vane-type film removal roller, the overall change trend of the area ratio first decreases significantly and then gradually changes more gently. The response surface changes greatly along the direction of the rotation speed of the vane-type film removal roller. In addition, at the zero level, the influence of the rotation speed of the vane-type film removal roller on the area ratio is more significant than that of the roller diameter on the area ratio.

The main reasons are as follows: the inclination angle of the blade is at the zero level, and the increase in the diameter of the roller leads to a decrease in the internal space of the device. With the increase in the rotating speed of the vane-type film removal roller, the internal air inflow of the device increases. Furthermore, under the combined action of the two factors, the internal pressure of the device increases, and the air inflow is sufficient, which leads to the gas gradually moving to both ends of the device. The difference in gas distribution at different positions decreases.

### 3.2.3. Parameter Optimization

Based on the processing conditions of the factory, this paper uses the optimization function of the Design-Expert software to optimize the regression model [48]. In the optimization process, the constraints on the test factors are as follows: the rotating speed

of the vane type film removal roller is 100 r/min~300 r/min, the inclination angle of the blade is 5°~25°, the diameter of the roller is 140 mm~219 mm; evaluation index: area ratio, takes the target minimum value. The final optimized parameter combination is as follows: the rotation speed of the vane-type film removal roller is 283 r/min, the inclination angle of the blade is 25°, the diameter of the roller is 219 mm, and the area ratio is 9.84%. Under the optimal parameter combination, velocity vector diagram of cross section 4 is shown in Figure 16.

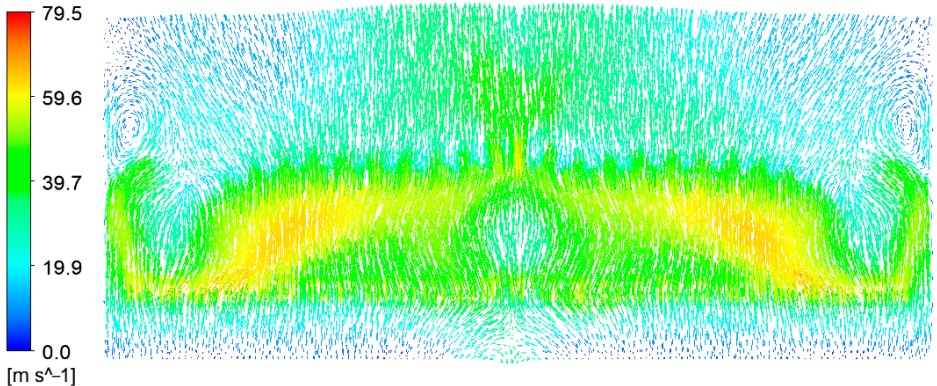

**Figure 16.** Velocity vector diagram of cross section 4 after optimization.

### 3.3. The Result of Test Verification

This paper selects the test field of Shihezi University (the test field is 4 m wide and 30 m long) as the test site. Using the abovementioned optimal parameter combination to process the film removal device and carries out the test based on the whole machine. During the test, the residual film picked up can be removed smoothly by a blade and blown into a residual film recovery box by the film removal device. The removed residual film is almost unaffected by the vortex. There is no secondary winding phenomenon between the removed residual film and blade, which verifies the accuracy of the simulation model. The test is shown in Figure 17, and the test results are shown in Table 6. Table 6 shows that the average film removal rate of the film removal device is 98.04%, which can meet the operation requirements of residual film recovery before sowing.

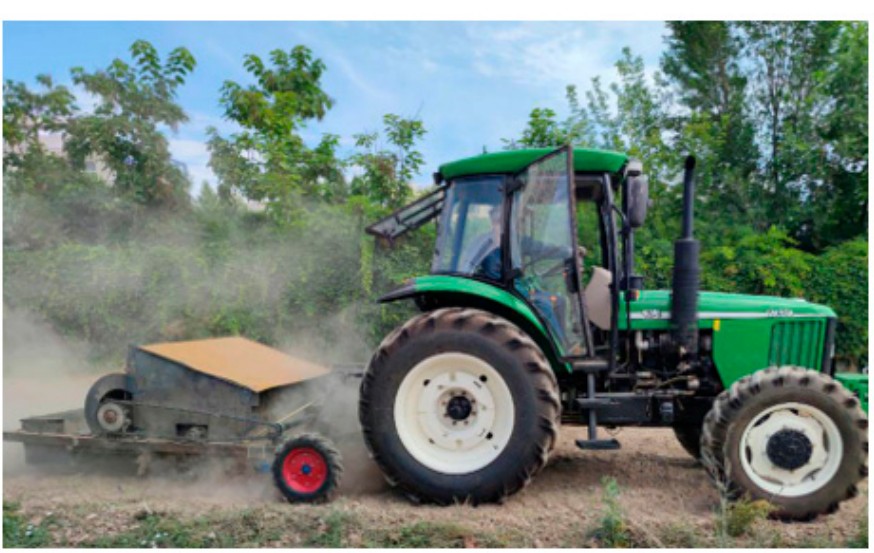

(**a**)

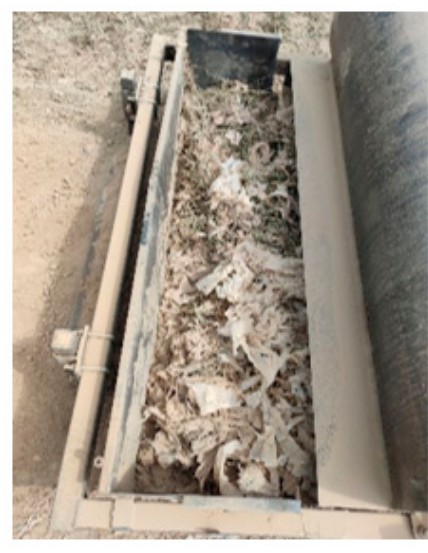

(**b**)

**Figure 17.** Field test (**a**) and removal and recovery effect of residual film (**b**).

**Table 6.** Test results.

| Test Number | Film Removal Rate/% |
|---|---|
| 1 | 97.15 |
| 2 | 98.20 |
| 3 | 98.78 |
| Mean value | 98.04 |

## 4. Discussion

This paper designed a "mechanical + airflow" combined film removal device by combining the advantages of pneumatic film removal and mechanical film removal. The device effectively solves the problem of the serious film wrapping phenomenon and poor film removal effect of the film removal device. Compared with the pneumatic film removal device, the design of the device eliminates the fan, and the film removal rate is higher (98.04%); compared with the telescopic rod tooth removal device, the device has simple structure and reliable performance; compared with the stripping blade type removal device, this device hardly exists film wrapping phenomenon. The research of this paper can provide some reference for the design of film removal devices in the future.

## 5. Conclusions

1. Addressing the problem of the serious film wrapping phenomenon and poor film removal effect of the film removal device of the residual film recovery machine, this paper designs a "mechanical + airflow" combined film removal device and analyses the structure and working principle of the film removal device. In addition, this paper also determines the important parameters of the film removal device and the conditions of film removal.

2. This paper uses the Fluent software to simulate the flow field inside the film removal device and obtains the important factors affecting the vortex region. Taking the rotation speed of the vane-type film removal roller, the inclination angle of the blade, the diameter of the roller as the test factors, and the area ratio of the vortex region to the effective region as the evaluation index, this paper designs a three-factor three-level orthogonal simulation test. Based on the processing conditions of the factory, this paper uses the optimization function of the Design-Expert software to optimize the regression model [32]. The final optimized parameter combination is as follows: the rotation speed of the vane-type film removal roller is 283 r/min, the inclination angle of the blade is 25°, the diameter of the roller is 219 mm, and the area ratio is 9.84%.

3. This paper uses the final optimized parameter combination to process the film removal device and carries out a field test based on the whole machine. The results of the field test show that the average film removal rate of the film removal device is 98.04%, and there is no wrapping phenomenon during the operation, which can meet the requirements of the recovery of the residual film before sowing.

**Author Contributions:** Conceptualization, S.X. and J.L.; methodology, S.X.; software, S.X.; validation, X.W. and Z.Z.; formal analysis, Z.Z.; investigation, S.X. and J.L.; resources, S.X.; data curation, S.X.; writing—original draft preparation, S.X.; writing—review and editing, S.X., X.C. and J.L.; project administration, J.L.; funding acquisition, J.L. All authors have read and agreed to the published version of the manuscript.

**Funding:** This research was funded by the Major Scientific and Technological Project of Bingtuan, grant number 2018AA001.

**Institutional Review Board Statement:** Not applicable.

**Informed Consent Statement:** Not applicable.

**Data Availability Statement:** All data are presented in this article in the form of figures and tables.

**Conflicts of Interest:** The authors declare no conflict of interest.

## Nomenclature

| | |
|---|---|
| $a$ | the maximum distance between the teeth of adjacent blades: mm. |
| $b$ | the tooth height of the blade, mm. |
| $c$ | the distance between adjacent teeth, mm. |
| $F_1$ | the force of the blade on the residual film when blades interact with . arc-shaped teeth |
| $F_2$ | the decomposition force of $F_1$ along the tip of the arc-shaped tooth, N. |
| $F_3$ | the decomposition force of $F_1$ along arc-shaped tooth's normal direction, N. |
| $\beta$ | the angle between $F_1$ and $F_3$, N. |
| $F_a$ | the air resistance of the residual film, N. |
| $F_\omega$ | the centrifugal force on the residual film, N. |
| $G$ | the gravity of the residual film, N. |
| $F_N$ | the supporting force of blade to residual film, N. |
| $F_f$ | the friction force of the residual film, N. |
| $\theta$ | the inclination angle of the detached blade, (°). |
| $\theta_1$ | the angle between the gravity of the residual film and the vertical direction of the detached blade, (°). |
| $c1$ | the resistance coefficient. |
| $p$ | the air density, 1.29 kg/m$^3$. |
| $s$ | the windward area of the residual film, m$^2$. |
| $v_1$ | the linear velocity of the blade, m/s. |
| $m$ | the mass of the residual film, kg. |
| $\mu$ | the static friction coefficient of the detached blade. |
| $r$ | the distance between the residual film and the rotating center of the vane-type film removal roller, m. |
| $v$ | the linear velocity of the arc-shaped tooth, m/s. |
| $L_1$ | the distance of film removal, m. |
| $F_b$ | the average force of the blade to the residual film on the arc-shaped tooth, N. |
| $R_1, R_2, R_4, R_5$ | curve radius of diversion shell, mm. |
| $R_3$ | the radius of roller, mm. |
| $L_2$ | the size of inlet, mm. |
| $L_3$ | the height of film conveying port, mm. |
| $L_4$ | the length of simplified model $L_4$, mm. |
| $L_5$ | the width of simplified model $L_5$, mm. |
| $L_6$ | The length of blade $L_6$. |
| $K$ | the film removal rate, %. |
| $m_1$ | the weight of residual film in the residual film recovery box, g. |
| $m_2$ | the quality of the residual film that is picked up but does not enter the residual film recovery box, g. |

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
