# Peer review of "Design of and Experiment on a Film Removal Device of an Arc-Toothed Residual Film Recovery Machine before Sowing"

_applsci, doi:10.3390/app11188551_

Round 1

Reviewer 1 Report

Review: applsci-1342325 Manuscript

Title: “Design of and experiment on a film removal device of an arc-toothed residual film recovery machine before sowing” by Shuaikang Xue et al

The research is very practical and the agricultural world can make use of studies of this type. I thank the researchers who have dedicated their time to this investigation and have also paid attention to the reader to illustrate the results and consequent choices.

Comments

  1. What kind of parameters the author considered for simulation and field tests need to be enlisted in one particular table. It would be meaningful for readers.
  2. The author mention materials for different parts. However, if you applied different kinds of materials and speeds, what will be the impact of the results? Need to clarify this.
  3. 2.3. Parameter optimization: Please describe more about the algorithm you used for optimization. A flow chart would be meaningful.
  4. Perhaps it would be worth quoting more of the world literature on film removal device. 35 items are not enough for this topic.
  5. It would be worth including a list with explanations of all notations used in the work at the beginning of the work.
  6. The work will be interesting if the comparison of theoretical results with the simulation results.
  7. I would also recommend reviewing the attached PDF file with suggestions and comments. 

Author Response

Response to Reviewer 1 Comments

The following is a point-to-point response to the reviewer’s comments. In addition, the revised manuscript has been revised according to the reviewer's comments, and the revised part has been marked in red. Special thanks for the reviewer’s professional suggestions. The comments not only promote the quality of the manuscript, but will play an important role in our later research work.

Point 1: What kind of parameters the author considered for simulation and field tests need to be enlisted in one particular table. It would be meaningful for readers.

Response 1: Thank you very much. For the simulation section, I have added a table: Table 1 and Figure 7. 2D schematic diagram of simplified model (Line 260-264, Page 9 in the revised manuscript); For field tests, I have added a table: Table 2 (Line 289-290, Page 11 in the revised manuscript)

Point 2: The author mention materials for different parts. However, if you applied different kinds of materials and speeds, what will be the impact of the results? Need to clarify this.

Response 2: Thank you very much. I have described the impact of different materials and speeds in the revised manuscript (Line 135-140, Page 5 in the revised manuscript; Line 234-246, Page 8 in the revised manuscript).

Point 3: 2.3. Parameter optimization: Please describe more about the algorithm you used for optimization. A flow chart would be meaningful.

Response 3: Thank you very much. According to your suggestion, I described that the algorithm used for optimization (Line 427-431, Page 17 in the revised manuscript). Supplementary notes: Optimization algorithm is an optimization function in Design export software. The optimization results can be obtained by adding constraints to the test factors during optimization. The specific process is shown in the figure below.

Point 4: Perhaps it would be worth quoting more of the world literature on film removal device. 35 items are not enough for this topic.

Response 4: Thank you very much. According to your suggestion, I have quoted 13 world literatures on film removal device in the revised manuscript. (Line 540-549, Line 562-563, Line 568-569, Line 574-576, Line 581-585, Line 605-612, Page 20-21 in the revised manuscript)

Supplementary notes: Dear professor, at present, there are relatively few literatures on the residual film stripping device. And some of the added literatures are about the whole machine, because these literatures about the whole machine include the research on the film removal device.

Point 5: It would be worth including a list with explanations of all notations used in the work at the beginning of the work.

Response 5: Thank you very much. According to your suggestion, I added a list with explanations of all notations used in the work. (Line 66-67, Page 2-3 in the revised manuscript)

Point 6: The work will be interesting if the comparison of theoretical results with the simulation results.

Response 6: Thank you very much. Under the optimal parameter combination, velocity vector diagram of section 4 is shown in Figure. 16. (Line 435-437, Page 17 in the revised manuscript). Through the comparison between Figure. 14 and Figure. 17, the optimized flow field has been well improved. Figure 17 verifies the correctness of the theoretical analysis.

Point 7: I would also recommend reviewing the attached PDF file with suggestions and comments.

Response 7: Thank you very much. I have revised the manuscript according to the suggestions in the PDF file. (in the revised manuscript)

PDF file with suggestions and comments.

Point 1: "1. Introduction": Authors highlight the research results in the available literature but in my opinion, the study of literature could be a little bit wider.

Response 1: Thank you very much. According to your suggestion, I have modified "1. Introduction" in the revised manuscript. (Line 40-48, Line 58-62, Page 1-2 in the revised manuscript)

Point 2: "pneumatic type, telescopic rod tooth type and stripping blade type [13-14]. ": Please add the reference individually.

Response 2: Thank you very much. According to your suggestion, I have modified it in the revised manuscript. (Line 55-57, Page 2 in the revised manuscript)

Point 3: "Mechanical + airflow" combined film removal device: Please add reference and description before about Mechanical + airflow" combined film removal device. (Line 73-74, Page 3 in the revised manuscript)

Response 3: Thank you very much. According to your suggestion, I added reference [25] and description before about Mechanical + airflow" combined film removal device. (Line 70-74, Page 3 in the revised manuscript)

Point 4: " Fluent software": Please describe why this software is important? (Line 77, Page 1 in the revised manuscript)

Response 4: Thank you very much. According to your suggestion, I have described the important of this software in the revised manuscript. (Line 77-78, Page 3 in the revised manuscript)

Point 5: " it contacts the high-speed rotating film stripping blades, and the film stripping blades have the effect of "scraping" and "hitting" the residual film to complete the film removal work.": Please add a reference. (Line 95-97, Page 3 in the revised manuscript)

Response 5: Thank you very much. According to your suggestion, I have added a reference in the revised manuscript. (Line 97, Page 3 in the revised manuscript)

Point 6: " a roller, film stripping blade (hereinafter referred to as blade), and blade connecting plate. The roller material is a seamless steel pipe with a certain wall thickness. The blade connecting plate is evenly welded on the surface of the roller. The main structure of the diversion shell is composed of the top cover, the bottom plate and the side plate. ": Please add the number from the figure. That readers can understand easily. (Line 106-113, Page 4 in the revised manuscript)

Response 6: Thank you very much. According to your suggestion, I have added the number from the figure. (Line 106-113, Page 4 in the revised manuscript)

Point 7: " which can achieve a good film removal effect and reduce the damage to arc-shaped teeth.": I think it's your hypothesis. Please rewrite this with more references. (Line 139-140, Page 5 in the revised manuscript)

Response 7: Thank you very much. According to your suggestion, I have rewritten this with more references. (Line 135-140, Page 5 in the revised manuscript)

Point 8: "2.3.1. Design of tooth profile of blade": You write well here. However, I recommend you, please write more clearly. Still, readers would be confused about the relationship between a, b and c. (Line 141, Page 5 in the revised manuscript)

Response 8: Thank you very much. According to your suggestion, I modified it more clearly in the revised manuscript. (Line 144-161, Page 5 in the revised manuscript)

Point 9: "2.4. Simulation model and setting of simulation parameters": If it was done than in what software, what type of elements, a figure with presented quality of mesh to check if velocity distribution is correct.

Response 9: Thank you very much. I added a description of the element type and I added a figure 6 named Section view of effect of mesh dividing. (Line 254, Line 258-260 , Page 9 in the revised manuscript)

Point 10: " Fluent software": Please Add software model, year and company. (Line 251-252, Page 9 in the revised manuscript)

Response 10: Thank you very much. According to your suggestion, I have changed " Fluent software" to ANSYS Fluent19.0 (Line 252, Page 9 in the revised manuscript)

Point 11: " Figure 6. Schematic diagram of simplified model.": Would you mind changing the mark using numbering in the figure?

Response 11: Thank you very much. According to your suggestion, I have changed the mark in the figure (figure 8, Line 265-268, Page 10 in the revised manuscript)

Point 12: " SIMPLE algorithm": how was applied algorithm? Need to add block diagram.

Response 12: Thank you very much. According to your suggestion, I added some descriptions for " SIMPLE algorithm" and added a reference [41] (figure 8, Line 275, Page 10 in the revised manuscript)

Supplementary notes: The" SIMPLE algorithm" is the setting of "The pressure-velocity coupling method" in the operation of fluent software. Through reading a lot of literature and materials, it shows that the algorithm is suitable for my model simulation.

Point 13: " Figure 7. Schematic diagram of cross-section selection": Would you mind changing the mark using numbering in the figure?

Response 13: Thank you very much. According to your suggestion, I have changed the mark in the figure 9 (figure 9, Line 305-307, Page 12 in the revised manuscript)

Point 14: " Test verification.": The validation test method needs to add to the materials and method section. afterward, you may discuss the results in the latter section.

Response 14: Thank you very much. According to your suggestion, I have added the validation test method to materials and method section (Line 280-300, Page 11 in the revised manuscript; Line 467-477,Page 18 in the revised manuscript)

Thank you very much for your valuable suggestions. I will work harder in my future research life.

Reviewer 2 Report

The problem associated with the use of film in agriculture is very great. At present, bodily degradable films are being introduced and used more and more. Returning to the manuscript, it is very important to look for new methods to recover the scrap film. The manuscript presents the structure of the machine and simulation of its functioning. This form of the manuscript may also be of value. The manuscript needs a few corrections. First of all, you should carefully check the MDPI requirements and adapt to them. In the abstract, the merits of the presented machine should be emphasized in more detail. Results and discussion should be written in saparate sections. Following the changes made, consideration may be given to the publication of the manuscript.

Author Response

Response to Reviewer 2 Comments

The following is a point-to-point response to the reviewer’s comments. In addition, the revised manuscript has been revised according to the reviewer's comments, and the revised part has been marked in red. Special thanks for the reviewer’s professional suggestions. The comments not only promote the quality of the manuscript, but will play an important role in our later research work.

Point 1: The problem associated with the use of film in agriculture is very great. At present, bodily degradable films are being introduced and used more and more. Returning to the manuscript, it is very important to look for new methods to recover the scrap film. The manuscript presents the structure of the machine and simulation of its functioning. This form of the manuscript may also be of value. The manuscript needs a few corrections. First of all, you should carefully check the MDPI requirements and adapt to them. In the abstract, the merits of the presented machine should be emphasized in more detail. Results and discussion should be written in saparate sections. Following the changes made, consideration may be given to the publication of the manuscript.

Response 1: Thank you very much. I have check the MDPI requirements and I added the description of the advantages of the machine and separated Results and discussion into different parts (Line 301, Page 11; Line 480-489, Page 19 in the revised manuscript)

Thank you very much for your valuable suggestions. I will work harder in my future research life.

Round 2

Reviewer 1 Report

The authors changed and revised the manuscript well.